# Virtual Intelligence: A Systematic Review of the Development of Neural Networks in Brain Simulation Units

**DOI:** 10.3390/brainsci12111552

**Published:** 2022-11-15

**Authors:** Jesús Gerardo Zavala Hernández, Liliana Ibeth Barbosa-Santillán

**Affiliations:** Department of Information Technology, Universidad de Guadalajara, Guadalajara 45100, Mexico

**Keywords:** computational architectures, brain functions, computational neuroscience, neural networks

## Abstract

The functioning of the brain has been a complex and enigmatic phenomenon. From the first approaches made by Descartes about this organism as the vehicle of the mind to contemporary studies that consider the brain as an organism with emergent activities of primary and higher order, this organism has been the object of continuous exploration. It has been possible to develop a more profound study of brain functions through imaging techniques, the implementation of digital platforms or simulators through different programming languages and the use of multiple processors to emulate the speed at which synaptic processes are executed in the brain. The use of various computational architectures raises innumerable questions about the possible scope of disciplines such as computational neurosciences in the study of the brain and the possibility of deep knowledge into different devices with the support that information technology (IT) brings. One of the main interests of cognitive science is the opportunity to develop human intelligence in a system or mechanism. This paper takes the principal articles of three databases oriented to computational sciences (EbscoHost Web, IEEE Xplore and Compendex Engineering Village) to understand the current objectives of neural networks in studying the brain. The possible use of this kind of technology is to develop artificial intelligence (AI) systems that can replicate more complex human brain tasks (such as those involving consciousness). The results show the principal findings in research and topics in developing studies about neural networks in computational neurosciences. One of the principal developments is the use of neural networks as the basis of much computational architecture using multiple techniques such as computational neuromorphic chips, MRI images and brain–computer interfaces (BCI) to enhance the capacity to simulate brain activities. This article aims to review and analyze those studies carried out on the development of different computational architectures that focus on affecting various brain activities through neural networks. The aim is to determine the orientation and the main lines of research on this topic and work in routes that allow interdisciplinary collaboration.

## 1. Introduction

Brain function studies have been highly relevant to different scientific disciplines for a long time. The sciences have focused on recognizing those epistemological aspects of brain function (how is knowledge generated?).

Throughout history, different thinkers have been interested in understanding how the brain works. Descartes offered the first approaches of relevance for these studies. A separation between the body and the mind was proposed, thus originating a dualistic problem in which cerebral activities belong to a world outside the material or human world. This approach to the mind at a purely ideological level is housed in theoretical assumptions about brain functioning since the mind is an immaterial entity without the possibility of experimentation “only through the subjective exercise of the knowledge: here and now”, Descartes’ vision brought an endless discussion of how the brain works, resulting in the classic “mind–brain problem”. This approach and the difficulty of avoiding the metaphysical character of the mind brings the rejection of the study of brain activities of a higher degree or involving aspects related to the use or presence of consciousness or conscious states.

Consequently, a distancing between neuroscientific studies and mental phenomena will continue for a long time (the second half of the 19th century and the first half of the 20th century). The main argument for this was the impossibility of scientific verification of the existence of consciousness within brain activities. Currently, science has not fully understood the problem of consciousness or conscious actions. The mind is an entity outside the physical world; the brain, on the other hand, is presented as an organism with a determined form and time. The reductionism of Penrose [1] or the functionalism of Baars [2] forced us to think about how the brain works. Studies in neuroscience have mainly focused their attention on those processes of a biological nature, which can be susceptible to measurement, experimentation and verification from any complexity that could question its formality as a science because the brain activities of a higher nature exceed scientific understanding. At present, different studies have made the explanation of mental life from observation, analysis and experimentation of the functioning of the brain possible through technology with the support of mechanisms and devices for the simulation, storage and processing of information to recognize the vast universe of the human mind.

The study of brain activities and IT participation in these phenomena have their antecedents in the second half of the 20th century. The discussion about high-order activities originated from studies by Alan Turing, where he proposed ways in which a machine or system can resemble human behaviour through algorithms and computation. Another of the most important principles was the one carried out by the mathematician John Von Neumann.

The possibility of similarity between the brain and a computer was affirmed, taking the brain functioning as a principle. The study of brain activities had been presented as a problematic phenomenon full of questions and enigmas. As stated by Ryle [3], Descartes’s categorical error brought the subjectivity of thought and its metaphysical character. The indubitableness of thought and the infallibility of our experience are possible because we are aware of these acts (we only superficially know what happens). That is, we only know the results of these processes, not the processes themselves. The neuroscientific community started to study brain functions, from knowing its biological behaviour [4] to its most complex mechanisms. Various ways of thinking have emerged to explain brain behaviour, such as Davidson’s anomalous monism [5], Searle’s biological naturalism [6], the dynamic core hypothesis of Tononi [7] and the interactionist dualism of Eccles [8]. The diversity of perspectives in the study of the brain and its functioning can be reduced to positions that start from the principal basis of understanding the study of brain phenomena: materialism and functionalism. However, as a consequence, this separation has produced constant discussion between the elements involved in the brain activities (mainly those related to consciousness) and how new models can be developed to explain the structure and behaviour of this type of activity scientifically [9].

The question of how the brain works must take into account different approaches considering how the brain works or behaves as follows:A.Series of causes, predispositions or internal properties in the subject’s mental states can be determined as intrinsic properties. Material descriptions seek the development of pure neuroscience away from problems concerning Penrose’s mind [1].B.Activities can be translated through the generation of programming languages that seek to interpret brain-behaviour blocks scientifically [10].C.Functional equality between a person and a digital computer which are in optimal conditions; that is, it is based on the solid assumption that these two entities perform the same abstraction or processing functions [11,12].

The explanatory principle of science is that brain activities and mental states arise biologically like any other human activity. Since it cannot be denied that this activity belongs to the physical world in the face of the findings presented by different techniques of brain measurement [13,14,15], which show the participation of certain brain parts in these activities or processes, the explanation of the functioning of consciousness is not simple. Different aspects such as the biochemistry presented during conscious activity or the generation of meta-representations about lived experiences [16] still represent an enigma that requires greater depth for computational studies in neuroscience.

However, this does not mean they should only reduce the explanation of consciousness or mental activities to their biological roots. Scientific advances show the possibility of exploring this enigmatic activity with new techniques and technologies that would make it possible to decode a vital part of this phenomenon [17,18]. One of the aims is to elaborate more specific explanations of the operation of this activity or emergent property of human life.

Many complex brain activities are derived from conscious experiences and the correlation between a neural system and its environment. These conscious activities, which occur in the subjects, are verifiable through the verbal manifestation of the issue experiencing these states and the detection of neuronal activity and plasticity through the synaptic exchanges derived from the action potentials executed [19] during a conscious brain process [20,21].

Conscious life on a biological level is presented through the stimuli generated from a subjective experience. These stimuli are defined through neural patterns of activation that determine the nature of the neural representation [22]. In turn, a neural model is translated as a code or language, which the brain uses to represent the information of the stimuli [23]. The task of analyzing this code is complex and can currently find its conceptualization within the quantitative descriptions offered by the nerve conduction equations of the computational models of experimental neuroscience [24,25].

Cognitive science development has brought with it the analysis and application of many faculties of the mind to mechanisms or devices [26]. Faculties such as vision, language or memory have been developed in “intelligent machines” [27,28]; however, these types of faculties are considered at a minor or primary level to the more complex activities of human nature. The development of these mental faculties in different machines has resulted from elaborating execution codes within computational languages. Their explanation finds meaning within them [29]. Currently, there are a series of questions about the possibilities that technology has to assimilate or become aware of an automaton mechanism [30].

Both systems (the mental and the computational) “process information” from their environment. Therefore, they are considered equivalent cognitive systems [18,31,32], assuming that two processing systems similarly execute certain tasks in such a way that they cannot be distinguished from one another; these could cause them to be considered the same or identical [33]. The task is focused on recognizing not the operations’ similarity between one system and another, but instead, the information method processed [34].

Computational models of the mind have developed an architecture in which they work by simulating the different processes controlled by different zones or areas of the brain, trying to reproduce these processes in the operation of a processor [35,36]. These mental processes synchronize and prioritize human activities working individually or in sync with other areas [37] which are found in some brain regions and act together in human activities. Most of the architectures in brain functions focus their attention on the primary elements of the brain. They have developed promising models, as in the case of neural networks [38,39,40].

What kind of digital architectures can be effective models for this task? The problem focuses on the difference in information processing and interpretation capabilities between an individual and a system.

Regarding the study of high-order brain activities or conscious human activities, a series of projects worldwide have focused on developing simulators and platforms that aim to execute certain brain activities. Currently, some models seek to simulate brain functions in real time and speed in synaptic processes, data management and the capacity to amalgamate certain brain activities related to consciousness, such as decision making.

Understanding the functioning of consciousness has become essential for neuroscientific research today. Therefore, it is necessary to consider multidisciplinary work as the basis of any approach to this organism (since we can understand biological behaviour). However, the subjective and phenomenological factors that make each individual a unique being continue to be incomprehensible to scientific understanding.

This paper aims to conduct a systematic review of different papers, their lines of action and their methods focused on studying brain function within computational neurosciences. The purpose is to recognize the main works on the development of brain simulators based on neural network architecture and to identify those aspects that can be studied in more detail in this line of research. The paper is organized as follows: Section 2 describes the construction of virtual intelligence: a systematic review of the development of neural networks in brain stimulation units and the performance metrics used to evaluate the model and presents the results of the different experiments, while Section 3 offers some discussion. Finally, Section 4 summarises the conclusions drawn from this work.

## 2. Methods and Results

The systematic review of the literature shows the results obtained in the search and classification of different papers aimed at developing this computational architecture used to stimulate brain function.

The present study found three main lines of research for the development of neural networks:Programming languages capable of simulating brain behaviour.Brain plasticity and learning.Processing and execution of cognitive tasks in real real time.

Some research has taken advantage of the development of imaging studies to gain insight into biological functioning and the different brain areas involved in cognitive processes. It has provided an excellent opportunity to develop computational architectures that aim to mimic the brain as precisely as possible within scientific studies in AI.

The investigations within the computational neurosciences have developed complex structures seeking to reconstruct the biological, anatomical and cognitive processes. They are present in this brain activity by developing diverse computational architectures. It interacts with their actions (neuromorphic computing, silicon chips, neural networks, multiprocessors and others).

### 2.1. Research Question and Objectives

At first, the establishment of the PICO question was proposed, which could offer a search route within this investigation to make it possible to trace the corresponding work methodology. Therefore, to carry out this systematic review, we started with the following research question:

What are the applications of neural networks in computational architectures for the study of brain functions?

At the time of identifying the object of study, the search terms and criteria used to answer the research question were obtained as follows, as shown in Figure 1:A.Identify the primary research into developing neural networks for brain stimulation of conscious activities.B.Determine the key concepts within the research question for the article search.C.Determine the exclusion or inclusion factors for the article search.

### 2.2. Inclusion and Exclusion Criteria

We carried out the review and analysis of three specialized databases in engineering and technology. The databases consulted are the following:IEEE Xplore (https://ieeexplore-ieee-org.wdg.biblio.udg.mx:8443/Xplore/home.jsp) (accessed on 12 August 2022);Computing Machinery Association (https://dl-acm-org.wdg.biblio.udg.mx:8443) (accessed on 23 August 2022);EBSCO Host, Library, Information Science and Technology Abstract (https://web-p-ebscohost-com.wdg.biblio.udg.mx:8443/ehost/search/basic?vid=0&sid=165ca839-ee96-4612-bc68-1d443e1073b7%40redis) (accessed on 29 August 2022).

Once the question was established, we focused on designing a search strategy to locate the most relevant information for this review. Specific exclusion criteria were needed to create the search strategy, such as the search criteria and the literary resources. It began by using the word neural network and awareness or brain as connectors as a criterion for search engines. From this, all academic articles written between 2017 and 2021 (full text) in English were selected, oriented to topics such as the brain, cognition, neural networks, learning, neurons, brain–computer interfaces, data analysis, machine learning, intelligence artificial, deep learning, meta-analysis, behaviour, neuroscience, communication and technology.

As shown in Table 1, a total of 3328 documents were found, organized as follows:

### 2.3. Data Extraction

The selection criteria for our research question were provided with greater clarity to identify relevant studies. The answer to the research question, as well as more specific exclusion criteria, took as criteria for the selection of publication topics oriented to research methodology, research, research funding, machine learning, brain, brain computational interfaces, artificial intelligence, cognition, meta-analysis, cognitive training, algorithms, data analysis, decision making, physiology brain, cognitive ability, deep learning and artificial neural network as shown in Table 2.

Finally, relevant articles through criteria, such as reviewing titles, summaries, or full texts of the works were selected. Studies focused on treatments or analysis of the behaviour of patients with diseases, brain medical studies, or anatomical research through neuro-imaging that were not oriented to the development of computational architectures were discarded.

In addition, we considered three brain study areas as reference points for computational architectures oriented to the simulation of higher-order brain processes. The areas taken as a decision pattern for the selection of the papers are highly relevant for studies in computational neuroscience because it is through them that the different architectures addressed in this study are implemented. The areas we refer to are the following: plasticity, memory and learning.

As a result, we obtained 21 academic articles that met the characteristics of the selection criteria.

The next step was to complete the reading of the selected works and extracting and synthesizing the recovered data. We used the quality assessment results to guide the interpretation of the review’s findings to indicate the types of computational architectures developed between 2017 and 2021 geared towards executing tasks similar to those of the human brain.

The data results are as follows:

Identify journals that published the papers, discuss the main lines of research in neural networks to the development of brain simulation units, report and discuss review findings according to the research questions and group them into areas. During the discussion, we interpreted the review results within the context of the research questions, in a broader context related to the current international interest in the brain. We provided some related works to support the findings.

### 2.4. Statistical Analysis

Initially, we grouped the information obtained by the type of journal and the number of publications collected. The exposure of information was carried out through the segmentation by percentages found in each of the types of journals indicated as show in Table 3.

We observed that 28.56% of the publications were found in 3 of the 21 reviewed journals (*IEEE Transactions on Biomedical Engineering, IEEE Transactions on Neural Networks and Learning Systems* and *Frontiers in Neurorobotics* with 9.52%, respectively); also, 71.44% of the remaining articles were distributed in one publication each for the 252 journals analysed (15 reports corresponding to 4.76% for each publication).

The development of different computational architectures with the use of neural networks has the task of mimicking brain functions as precisely as possible. Having the rational capacity that humans show has been one of the main objectives for studies in computational neuroscience. Knowledge of forms of learning has its primary basis in the information inputs that come from the experience of individuals with the environment. To acquire this knowledge, they need input channels through which the information reaches the brain. The main input channels for humans are the senses, which receive information and transform it into input data of different characteristics.

As shown in Table 4, the input channels found in the selected papers are the following:

The input data that predominate in the development of computational architectures for the simulation of brain functions come from visual stimuli, with 57.12% of the publications. In second place are mixed architectures, which take input data from the environment (visual or language [hearing or speech]) and use principles of neural connection, such as plasticity or neuronal synapses, with 23.8%, while the input data collected from language (hearing or speech) are in last place with 9.52%. The analysis found only two publications of theoretical research unrelated to experimentation, comprising 9.52%.

The data show that the primary input data for developing computational architectures for the simulation of brain functions are derived from visual information. This tendency is due to the ability to translate data from visual stimuli into machine language.

Computational neurosciences reference the senses as input channels through which the information reaches a processing centre. However, it has not been possible to imitate the senses for the transcription of data that AI can transform into knowledge.

The review found that the use of neural networks to develop computational architectures is oriented toward the design of the networks, followed by learning algorithms to simulate different brain functions in 38.1% [41,42,43,44,45,46,47,48]. Next, the development of brain simulation software is 14.3% [49,50,51], and the development of hybrid architectures (using brain computing interfaces supported by neuromorphic processors) accounts for 14.3% [52,53,54]. The development and improvement of brain computing interfaces was 9.5% [55,56], as was analysis and database storage through machine learning [57,58]. Finally, 4.8% was shown for the design of neuromorphic processors [59]. The remaining 9.5% of the selected publications are critical reviews on the development of AI in brain stimulation through theoretical models [60,61].

The computational architectures developed in the reviewed publications are shown in Table 5:

Finally, we found four application areas in which neural networks work to develop computational architectures oriented to the simulation of brain functions. These areas or units were classified as follows:New generation data collection for the study of brain activities.Reproduction of the brain structure and its primary functions through the development of brain simulation software to study molecular and subcellular aspects and neuronal functioning at a micro- and macroscopic level is shown.Development of cognitive computing through neuromorphic processors and silicon chips.The design of brain models for executing brain tasks, such as behaviour, decision making and learning.

The parameter used to establish paper selection measures the degree of relevance on a scale of 0 to 10, where 0 is considered unrelated, and 10 is regarded as a complete relationship. Figure 2 shows the results obtained from the analysis carried out regarding the relationship of the papers with the degree of relevance in each area.

## 3. Discussion

This systematic review shows the main studies that dealt with the development of computational architectures designed to simulate brain function between 2017 and 2021, as shown in Figure 3. We observed that the principal applications of these studies are directly related to developing cognitive processes.

The architectures found are strictly logical which considers the possibility of structuring architectures that may be capable of simulating some of the most complex human brain activities or activities related to consciousness.

One of the essential aspects of our review was the low prevalence of studies in this area in the 3328 publications we searched. Only 21 of them studied this phenomenon which means that only 0.63% of the comprehensive studies on neural networks were directed to this research area, according to the data presented (2.75% of the 762 studies found with specific criteria). It should be noted that most of the publications were directed to the development of the design of neural network structures for medical use (learning through databases for detection of diseases such as cancer, Parkinson’s, Alzheimer’s, among others).

One of the principal challenges found in this study is the complexity of the multiple data inputs of the human being and how these inputs are translated into information for the brain. Today, the main data inputs for this type of architecture come mainly from visual or language stimuli that can be translated through binary language. However, there is still much to be done in the case of other data inputs for the human (touch, taste, smell) that allow a better understanding of the environment and, therefore, better performance.

Furthermore, one of the main challenges of computational neuroscience is to design components or architectures based on the human structure and its functioning. Currently, the construction of computational elements with characteristics that can more accurately simulate brain processes is proposed, from basic biological functions to complex brain functions (development of sensors and sensory infrastructure, neural networks with the ability to simulate chemical and electrical exchanges, synaptic capacity, sensory motor strategies for control and cognition and knowledge transfer and diffusion), all of this responding to the initial question of our research: how is knowledge generated?

This systematic review found specific lines of research in recent works and publications as show in Figure 4. The application of neural networks is the main structure for generating computational architectures for the development and simulation of brain functions. These works mainly focused on the application of computational architectures for the study of the following functions:A.Memory.B.Learning.C.Decision making or free will.

We observed that the main ways for the collection, handling and storage of input data and for the implementation and development of computational architectures based on neural networks are the following:Recognition, classification, memory and data deduction through inputs through visuals.Acoustic or speech channels that are processed through electrical action potentials.Hybrid architectures that have neuromorphic processors with better capacities and speeds.Algorithms and new forms of synaptic exchanges based on chemical processes.

Faced with these challenges, international projects and initiatives have been created that aim to generate a brain and neuroscientific research network that contributes and disseminates the results obtained from its scientific practice. All this is from a multidisciplinary perspective using top-level techniques that allow the observation and measurement of phenomena related to the brain and its functioning.

Within these projects, lines of research were identified that study brain functions and the areas of the brain that participate in these processes through different investigations. The principal research projects focus on the collection, management and dissemination of data at an international level in neuroscience, brain simulation, theoretical approaches and multiscale models of the brain, brain computing focused on data analysis and robotics and imaging techniques for brain mapping and observation of brain processes.

Among the main initiatives, we can point out the following:The Human Brain Project ( HBP ) is a collaborative European research project with a ten-year program launched by the European Commission’s Future and Emerging Technologies in 2017 (https://www.humanbrainproject.eu) (accessed on 5 August 2022).The Brain Initiative was developed in the United States in 2014. It has the support of several government and private organizations within the United States but is under the direction and financing of the NIH (https://braininitiative.nih.gov) (accessed on 12 September 2022).Center for Research and Cognition in Neuroscience includes the neuroscience research laboratory founded in 2012 as a research center of the faculty of psychological science and education of the Université Libre de Bruxelles (https://crcn.ulb.ac.be) (accessed on 7 September 2022).

The objective of these initiatives is to design new methods and technologies for the study of the human brain, promote new models and images for brain exploration by studying the behaviour of individual cells and the interaction with neural circuits in time and space, create and operate scientific research structures for brain research, cognitive neuroscience and other scientific disciplines related to neuroscientific studies. The results of their work are available through their websites through different resources: journal publications, scientific achievements and deliverables by project phase.

The results obtained from our work and the analysis of recently published literature have shown that current works focus on developing one or more joint applications for the development of computational architectures. In addition, it should be noted that we identified six lines of research that verify these findings during our systematic review. Within the lines of research for the area of computational neuroscience, the following stand out:A.Large-scale recording and modulation in the nervous system.B.Next generation brain imaging.C.Integrated approaches to brain circuit analysis.D.Neuromorphic computing with biological neural networks as analog or digital copies in neurological circuits.E.Brain modeling and simulation.

These findings can agree with the results obtained in this systematic review on the main applications of neural networks in the development of computational architectures (Figure 5).

It is worth mentioning that the projects and initiatives mentioned are investigations that work in medium and long-term objectives and show their results through publications within their platforms where the main advances in their different research areas can observe in their platforms. These initiatives stand out among the most important projects around the world for their vision and multidisciplinary and collaborative work with international organizations such as universities, health systems and research centers.

## 4. Conclusions

The use and application of neural networks in the development of computational architectures for the execution of tasks that involve the human brain is undoubtedly a growing field of research. Currently, computational neuroscience studies have managed to imitate functions of a rational nature for the human mind. However, multiple factors still keep more complex brain tasks at a distance.

Advances in computational neuroscience studies have proven very useful in understanding new unknowns in brain behaviour. Consequently, multiple types of computational architectures are applying this new knowledge in the different areas of IT.

Applying different technologies is a fundamental step in studies of the brain and knowledge of its functioning. This knowledge allows the development of many types of computational architectures based on a structure close to that of brain function. Neural networks have made it possible to simulate many logical processes of the brain and how a neuron can behave in order to develop devices with a specific capacity or type of intelligence.

As mentioned in the previous section, different initiatives are working to discover new pathways towards understanding the brain. It is extremely important to delve into the work from these initiatives after this study, since many of their works are still under investigation.

Although advances in neuroscience have made it possible to delve into the functioning of brain processes at a biological and chemical level, human behaviour cannot be the object of biological or physical reductionism in this area. However, scientific advances have shown that achieving a complex and holistic understanding of brain function requires a multidisciplinary vision, which must start from the basic physical foundations of this process and be supported by emerging elements that technology could offer.

Knowledge of brain behaviour is necessary for understanding human behaviour, but this type of knowledge can also be of vital importance for issues related to the prevention and treatment of brain diseases,.

Finally, the purpose of examining each of these approaches is the need to offer a general theoretical vision of how studies in this area have been investigated and to point out their main discussions. It should be noted that the approaches offered are a way of showing what the main objectives around this topic have been. Within the main tasks for the study and development of research in this field of computational neuroscience we can point out:The need to build multidisciplinary projects for the study of brain functions.The application of multiple technologies for the simulation and modeling of brain activities.The creation of a global platform of free access for researchers.

However, computational neuroscience studies have as their main challenge one of the most enigmatic parts of the brain, consciousness. Knowledge of the brain functions that involve consciousness and the different areas of the brain that participate in this process is one of the main objectives of these disciplines. The possibility of developing devices capable of recognizing themselves and interacting with their environment is one of the most critical challenges for research in these disciplines and for future developments in AI.

## Figures and Tables

**Figure 1 brainsci-12-01552-f001:**
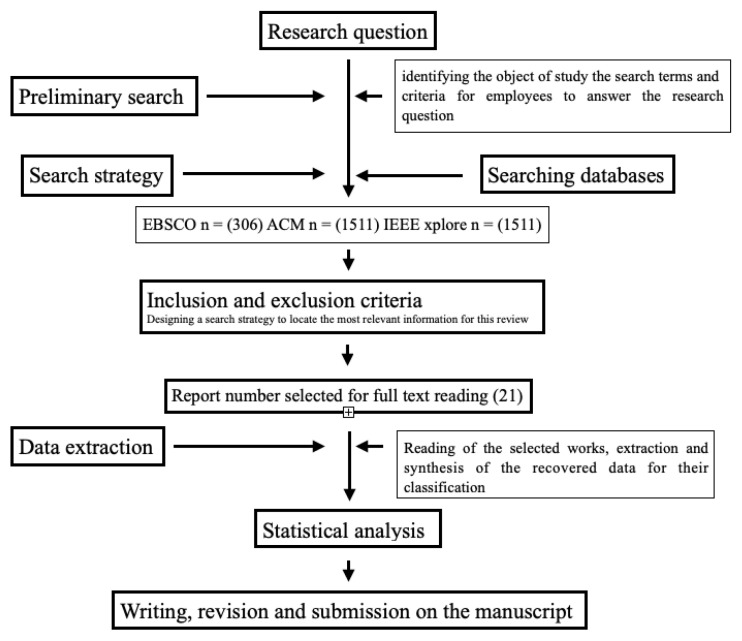
The flowchart shows the selection criteria made.

**Figure 2 brainsci-12-01552-f002:**
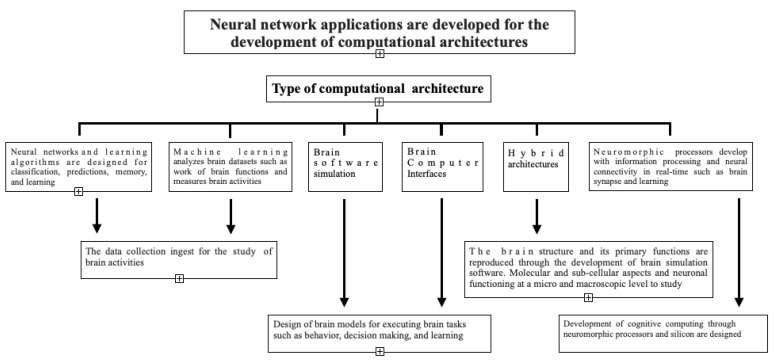
Neural networks applications and the main areas in which they are apply.

**Figure 3 brainsci-12-01552-f003:**
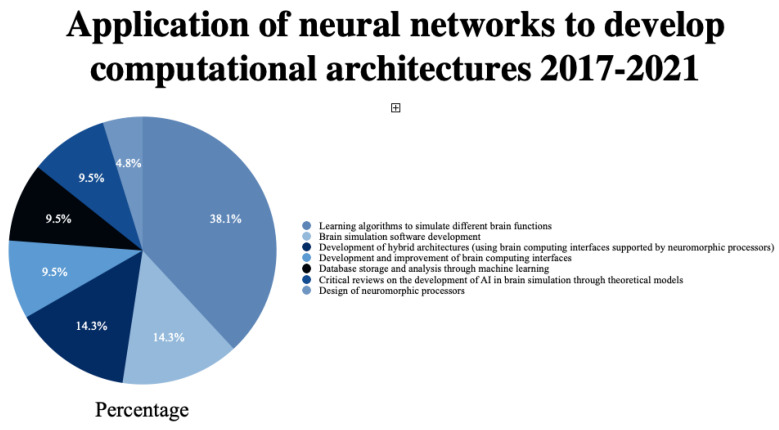
Main types of computational architectures developed through the application of neural networks to simulate brain tasks.

**Figure 4 brainsci-12-01552-f004:**
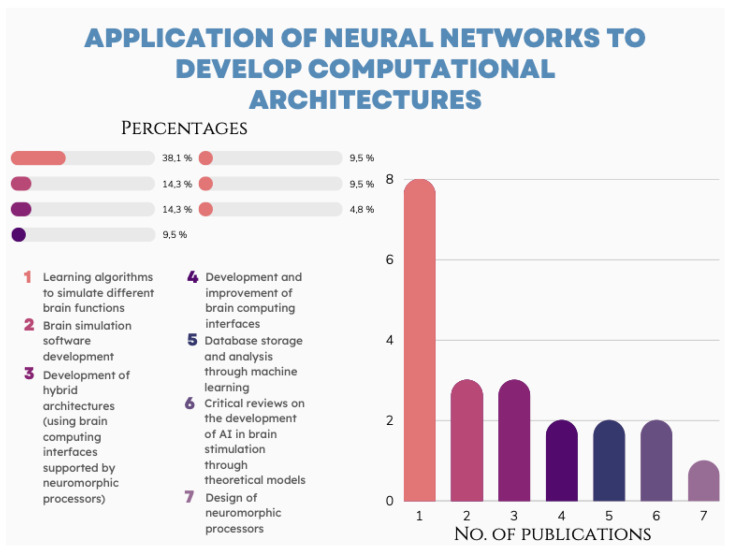
The results of the systematic review of the 21 publications show the main applications for neural networks in the development of computational architectures (7) and the number of studies oriented to each application.

**Figure 5 brainsci-12-01552-f005:**
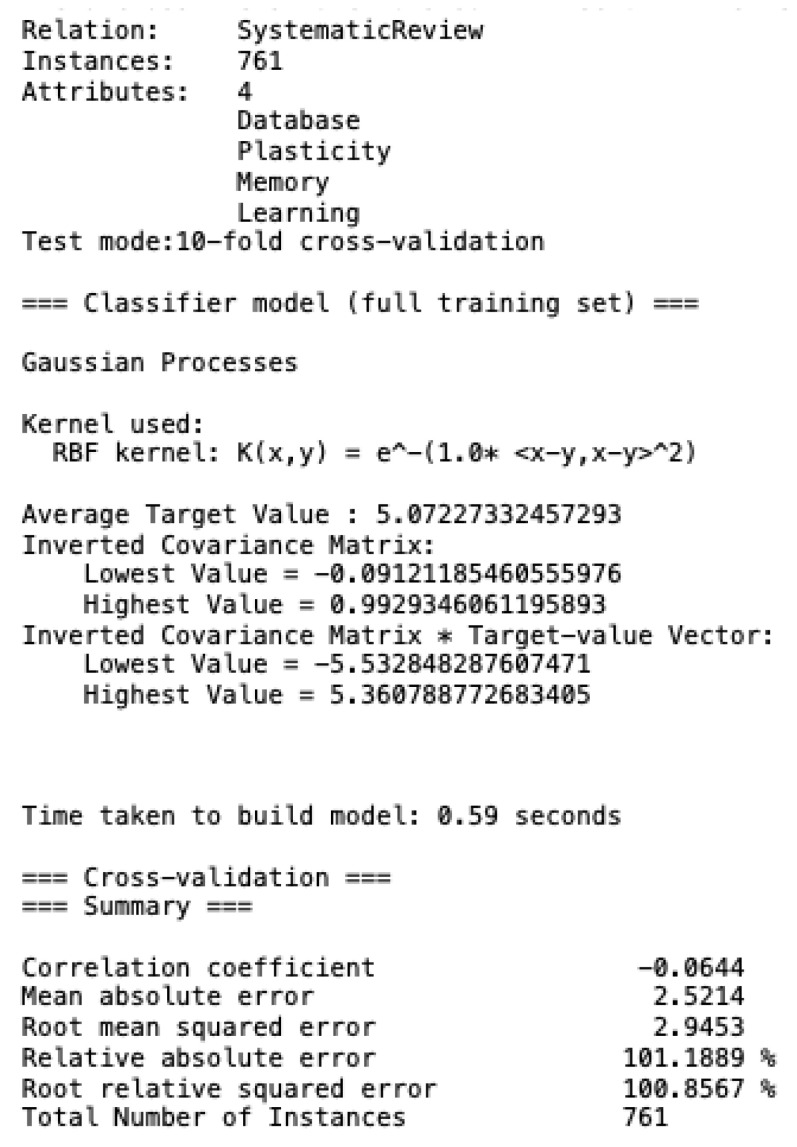
Gaussian processes for 761 works in three clusters: plasticity, memory and learning.

**Table 1 brainsci-12-01552-t001:** Total publications found in the database.

Database	Publication
ACM	1511
EBSCO	306
IEEE	1511

**Table 2 brainsci-12-01552-t002:** The number of publications results after the application of exclusion criteria.

Database	Publication
ACM	369
EBSCO	115
IEEE	278

**Table 3 brainsci-12-01552-t003:** Total of publications classified by type of journal.

Journal	N. Paper	Title	Types of Computational Architecture
*Biological Cybernetics*	1	A neural model of schemas and memory encoding	Design of neural networks and learning algorithms for classification, predictions, memory, and learning
*Computer*	1	Biologically driven artificial intelligence	A critical review of the development of AI in brain stimulation through theoretical models
*Connection Science*	1	Interactive natural language acquisition in a multi-modal recurrent neural architecture	Design of neural networks and learning algorithms for classification, predictions, memory, and learning
*Frontiers in Computational Neuroscience*	1	The neuroscience of spatial navigation and the relationship to artificial intelligence	Machine learning to analyze brain datasets (work of brain functions, measure of brain activities)
*Frontiers in Neurorobotics*	2	From near-optimal Bayesian integration to neuromorphic hardware: a neural network model of multisensory integration—a brain-inspired model of theory of mind	Design of neural networks and learning algorithms for classification, predictions, memory, and learning—A critical review of the development of AI in brain stimulation through theoretical models
*IEEE Access*	1	Study of recall time of associative memory in a memristive hopfield neural network	design of neural networks and learning algorithms for classification, predictions, memory, and learning
*International Journal of Advanced Robotic Systems*	1	A noninvasive brain–computer interface approach for predicting motion intention of activities of daily living tasks for an upper-limb wearable robot	Brain–computer interfaces
*IEEE Journal of Biomedical and Health Informatics*	1	Learning discriminative spatiospectral features of erps for accurate brain-computer interfaces	Brain–computer interfaces
*IEEE Journal on Exploratory Solid-State Computational Devices and Circuits*	1	Subthreshold spintronic stochastic spiking neural networks with probabilistic hebbian plasticity and homeostasis	Design of neural networks and learning algorithms for classification, predictions, memory, and learning
*IEEE Journal of Translational Engineering in Health and Medicine*	1	Integrated development environment for eeg-driven cognitive-neuropsychological research	Brain software simulation
*IEEE Transactions on Biomedical Engineering*	2	Modeling hierarchical brain networks via volumetric sparse deep belief network (VS-DBN)—feasibility of automatic error detect-and-undo system in human intracortical brain-computer interfaces	Design of neural networks and learning algorithms for classification, predictions, memory, and learning—Brain software simulation
*IEEE Transactions on Cognitive and Developmental Systems*	1	DAC-h3: A proactive robot cognitive architecture to acquire and express knowledge about the world and the self	Hybrid architectures
*IEEE Transactions on Computer-Aided Design of Integrated Circuits and Systems*	1	A compact gated-synapse model for neuromorphic circuits	Hybrid architectures
*IEEE Transactions on Neural Networks and Learning Systems*	2	Dendritic neuron model with effective learning algorithms for classification, approximation, and prediction—a brain-inspired framework for evolutionary artificial general intelligence	Design of neural networks and learning algorithms for classification, predictions, memory, and learning—Hybrid architectures
*IEEE Transactions on Neural Systems and Rehabilitation Engineering*	1	A real-time movement artifact removal method for ambulatory brain-computer interfaces	Machine learning to analyze brain datasets (work of brain functions, measure of brain activities)
*IEEE Transactions on Pattern Analysis and Machine Intelligence*	1	SimiNet: A novel method for quantifying brain network similarity	Design of neural networks and learning algorithms for classification, predictions, memory, and learning
*Neural Computation*	1	Controlling complexity of cerebral cortex simulations—I: CxSystem, a flexible cortical simulation framework	Brain software simulation
*Proceedings of the IEEE*	1	Advancing Neuromorphic Computing With Loihi: A Survey of Results and Outlook	Development of neuromorphic processors (information processing, neural connectivity in real time [brain synapse], and learning)

**Table 4 brainsci-12-01552-t004:** Types of inputs found in the selected papers.

Input Data	Paper Related
Inputs data through visual datasets or experience in real time	12
Inputs data through language datasets or experience in real time	2
Others (mixes inputs data and neural connection)	5

**Table 5 brainsci-12-01552-t005:** Computational architectures in selected publications.

Database	Publication
Design of neural networks and learning algorithms for classification, predictions, memory, and learning.	8
Brain software simulation	3
Hybrid architectures	3
Machine learning to analyze brain datasets (work of brain functions, measure of brain activities)	2
Brain–computer interfaces	2
Development of neuromorphic processors (information processing, neural connectivity in real time [brain synapse], and learning)	1
A critical review of the development of AI in brain stimulation through theoretical models	2

## Data Availability

Not applicable.

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
