# Peer review of "Virtual Intelligence: A Systematic Review of the Development of Neural Networks in Brain Simulation Units"

_brainsci, 2022, doi:10.3390/brainsci12111552_

Round 1
Reviewer 1 Report
In this review, the authors, using the principal articles of three databases oriented to computational sciences, analyze multiple kinds of architecture developed, to know the current objectives of neural networks in studying the brain.
I have enjoyed reading this manuscript and I found it very interesting and, even if I’m not an expert on the technical issues related to literature reviews, to me, the work seems well performed and sufficiently detailed. The figures are really well done and clear, and the discussion is complete.
Author Response
Thank you for your comments. We made changes to make this review more complete.
Reviewer 2 Report
This bibliometric analysis seems to aptly depict the scenario in development of brain simulation units. The review does a great job in statistically analyzing the published literature. But there is a major concern. The authors have already mentioned in the discussion section that the number of articles considered, after a sheer screening process, are too low i.e. 21. I truly agree with this fact. This makes their conclusions somewhat erroneous. For example, 1 article correspond to 4.76%. So, deriving conclusions based on 1 or 2 articles for a category seems erroneous. Thus, my suggestion would be to consider publications from last 5 years or 7 years or 10 years whichever seems feasible. The other suggestions are mentioned below:
11. The manuscript can be enriched by providing a flowchart for the methodology part which would depict how screening was performed step by step.
2 2. How does research funding matter as an exclusion criterion for selecting articles? Please explain?
33. The manuscript would greatly benefit by making Table 3 more elaborate. For each journal kindly mention the titles of the articles, the year of publication and types of computational architecture mentioned in the article, if possible.
4 4. The authors can consider pubmed as an important source for article collection because there are also many biology related journals that has good articles based on the present study.
55. In line 235, the author mentions “How the data is presented firstly:”. Instead of this the authors can write “The steps followed for presentation of the data” and then mention the points either as bullet points or numbers.
66. Pg 5, Line 204 – “Computing” instead of “omputing”.
77. Line 249, use % sign instead of percent.
88. Line 250 is not clear. Which are those 3 journals? Kindly mention the name of the journals correctly.
99. Table 4 heading – Replace “find” with “found”.
110. The venn diagram in figure 3 shows no overlapping regions so it seems unnecessary.
Author Response
We consider your observation of great interest actually we have a mistake and the period of time is between 2017-2021. However, there is a lot of information on brain research and most of them are used for the knowledge and observation of other processes or states of the brain and the study of the development of brain diseases and not for the concrete study of computational architectures using neural networks and focus on high brain activities. In this review, we can observe and highlight the lack of studies focused on the knowledge of more complex activities of the human brain and the need for a multidisciplinary approach. The review is aimed at pointing out the need to be able to work on technological projects in this area since the result of future research can be beneficial for different areas of human life. We understand that we could offer a more extensive selection for length or years as mentioned, and your suggestion could open future studies on this topic than as you indicate. Thanks.
1.1 We include it.
2.2 Our research organization's project financing did not influence the exclusion criteria. However, it should mention that access the databases through the bibliographic resources available to the University of Guadalajara.
3.3 We work the data and only not include the years because the study mention it.
4.4 We carried out a query within the PubMed database and found that there were 126 articles under the search criteria mentioned in the paper. However, when making a more precise selection, none of them talk about the development of a type of computational architecture that focuses on studying brain activities involved with consciousness or of a higher degree. Most of the studies found focus on the analysis related to the study of brain diseases, which is part of our exclusion criteria.
5.5 The change was made it.
6.6 The change was made it.
7.7 The change was made it.
8.8 The change was made it.
9.9 The change was made it.
110. We consider your observation and delete the Venn diagram
Reviewer 3 Report
This manuscript is understood as review article dealing with computational architectures simulating human intelligence. Based on papers of three computational sciences databases (EbscoHost Web, IEEE Xplore, and Compendex Engineering Village), the authors identify and discuss current research objectives of neural networks in studying the brain.
While the used methodolgy is quite interesting and worth contributing to an international paper, the manuscript lacks a strong and important point of (the nature of) a review article: Neither the discussion nor the the conclusion section presents a targeted summary and points to open research fileds which are connected to established leading publications (studies) of the community. I would expect that the authors would discuss identified open research topics in connection to recently published (2020-2022) literature. Without this linkage, this review would remain a solitary piece in the international pool of publications. As their approach is good and the topic important, the authors should definitely be given a chance to revise their manuscript to better present and strengthen the results of this paper.
Author Response
Thank you for your comments. We have worked on your observations and included information on points to open research files connected to established leading publications (studies) from the community and identified open research topics related to recently published literature (2020-2022), as you mentioned.
Reviewer 4 Report
Report on manuscript
Virtual intelligence: a systematic review of the development of neural networks in brain simulation units.
by
Jesús Gerardo Zavala Hernández, Liliana Ibeth Barbosa-Santillán
· Overview of manuscript
The authors presented a systematic review that analyses multiple kinds of architecture that have developed, taking the fundamental structure of the brain to recognize the main works on the development of brain simulators based on neural network architecture and to identify those aspects that can be studied in more detail in this line of research.
· Comments on text
1. English
The English in this paper is good. But needs some corrections such as "the neuron." in abstract, and few typos.
Comments
I highly recommend improving statistical analysis section. This section is almost the main section but unfortunately lack of statistical analysis made it less important.
Author Response
Response to Reviewer 4 Comments Thank you very much for your comments:
We have worked on the points you point out:
POINT 1.English
The English in this paper is good. But needs some corrections such as "the neuron." in abstract, and few typos.
Response 1: Please provide your response for Point 1
1. We review the manuscript to verify typos, grammar errors, and corrections
in expressions that improve the style.
POINT 2.
I highly recommend improving statistical analysis section. This section is almost the main section but unfortunately lack of statistical analysis made it less important.
Response 2: Please provide your response for Point 2.
2. We include the Gaussian Processes that work with three clusters, plasticity, memory, and learning, to show the relevance of the selected papers for the objectives of the studies of brain processes. It can be seen in figure 4, located on p. 12.

Round 2
Reviewer 2 Report
The authors have successfully adressed all the queries.
Author Response
Thank you for your contributions to our work.

Reviewer 3 Report
The authors re-submitted a revised manuscript. In their response letter, they write about changes made in the manuscript regarding open research questions, with references from the last three years. The response letter remains without any further details from the text, and, unfortunately, I could not identify any change made in the discussion or conclusion section. Against this background, I recommend the rejection of this manuscript.
Author Response
Thank you for your comments. We will respond regarding the requirements that you made in the first round and that we seek to respond to:
"The response letter remains without any further details from the text, and, unfortunately, I could not identify any change made in the discussion or conclusion section".
Point 1: Neither the discussion nor the conclusion section present a specific summary and points to open research files that are connected to established leading publications (studies) in the community.
Response 1:
We have worked to detail a specific summary and points to open research files connected to the community's established leading publications (studies).
We have detailed the primary axes, the main ways for the collection, handling, and storage of input data, for the implementation and development of computational architectures based on neural networks for the development and simulation of brain functions in our research and recent studies (page 13 lines 327-344).
Besides, we had included how this research is being addressed by international projects of great importance such as the Human brain project, the BRAIN initiative, and laboratories such as the CRNC (p. 13 lines 345-366)
Point 2: I expect the authors to discuss the identified open research topics in relation to the recently published literature (2020-2022).
Response 2:
We also pointed out the main lines of research that have been opened within these projects (through its publications and scientific achievements) and how they are directly related to the types of applications found in this review (page 13, lines 375-395).
Work has been done to point out the main challenges faced by this type of research in the development of new technologies that focus both on the biological nature of brain function and the need to generate process simulations of its capabilities—higher-order cognition (Conclusions). Finally, mention is made of the importance of creating a collegiate and multidisciplinary network for the development of future research.

Reviewer 4 Report
The authors have addressed the comments.